# Improving short-term sea ice concentration forecasts using deep learning

Cyril Palerme[1], Thomas Lavergne[1], Jozef Rusin[1], Arne Melsom[1], Julien Brajard[2], Are Frode Kvanum[1], Atle Macdonald Sørensen[1], Laurent Bertino[2], and Malte Müller[1]

[1]Norwegian Meteorological Institute, Oslo, Norway
[2]Nansen Environmental and Remote Sensing Center, Bergen, Norway

**Correspondence:** Cyril Palerme (cyril.palerme@met.no)

**Abstract.** Reliable short-term sea ice forecasts are needed to support maritime operations in polar regions. While sea ice forecasts produced by physical-based models still have limited accuracy, statistical post-processing techniques can be applied to reduce forecast errors. In this study, post-processing methods based on supervised machine learning have been developed for improving the skill of sea ice concentration forecasts from the TOPAZ4 prediction system for lead times from 1 to 10 days. The deep learning models use predictors from TOPAZ4 sea ice forecasts, weather forecasts, and sea ice concentration observations. Predicting the sea ice concentration for the next 10 days takes about 4 minutes (including data preparation), which is reasonable in an operational context. On average, the forecasts from the deep learning models have a root mean square error 41 % lower than TOPAZ4 forecasts, and 29 % lower than forecasts based on persistence of sea ice concentration observations. They also significantly improve the forecasts for the location of the ice edges, with similar improvements as for the root mean square error. Furthermore, the impact of different type of predictors (observations, sea ice and weather forecasts) on the predictions has been evaluated. Sea ice observations are the most important type of predictors, and the weather forecasts have a much stronger impact on the predictions than sea ice forecasts.

## 1 Introduction

Due to increasing maritime traffic in the Arctic (Gunnarsson, 2021; Müller et al., 2023), there is a growing demand for reliable short-term sea-ice forecasts that can support marine operations (Wagner et al., 2020). While short-term sea-ice forecasts are operationally produced by several institutions using dynamical models (e.g. Sakov et al., 2012; Smith et al., 2016; Barton et al., 2021; Williams et al., 2021; Ponsoni et al., 2023; Röhrs et al., 2023), the usefulness of these forecasts in Arctic navigation is often limited by their inaccuracies (Veland et al., 2021). Melsom et al. (2019) reported that the location of the ice edge is predicted with a mean accuracy of 39 km in 5-day forecasts from the TOPAZ4 prediction system (Sakov et al., 2012), with larger errors during the summer when most of the maritime traffic occurs (Müller et al., 2023). Furthermore, the sea ice concentration (SIC) forecasts from the regional model Barents-2.5km v2.0 are, in most cases, not better than persistence of SIC observations for short lead times (Röhrs et al., 2023).

It is common practice to post-process weather forecasts produced by dynamical (physical-based) models in order to improve their skill. Statistical correction techniques have been applied to atmospheric forecasts at time scales ranging from hours to seasons (e.g. Wang et al., 2019; Vannitsem et al., 2021; Frnda et al., 2022; Roberts et al., 2023), particularly on essential variables for end-users such as temperature, wind, and precipitation. In sea ice forecasting, most post-processing methods have been developed for subseasonal to seasonal time scales (e.g. Zhao et al., 2020; Director et al., 2021; Dirkson et al., 2019, 2022), but short-term sea ice forecasts produced by dynamical models are usually not post-processed despite their potential interests for end-users (Wagner et al., 2020). Nevertheless, Palerme and Müller (2021) showed that the errors of short-term sea ice drift forecasts (up to 10 days) from the TOPAZ4 prediction system (Sakov et al., 2012) can be significantly reduced using random forest models (by 8 % and 7 % for the direction and speed of sea ice drift, respectively). These post-processed sea ice drift forecasts have been distributed on the IcySea commercial application from 2020 to 2024 (https://driftnoise.com/icysea.html ; von Schuckmann et al., 2021), and can be considered as an exception in operational short-term sea ice forecasting.

Another approach consists of developing statistical sea ice forecasts without using dynamical sea ice model outputs. This has been used for sea ice forecasting at different time scales (e.g. Kim et al., 2020; Fritzner et al., 2020; Liu et al., 2021; Andersson et al., 2021; Grigoryev et al., 2022; Ren et al., 2022), with the advantage of greatly reducing the computational cost compared to dynamical models. Andersson et al. (2021) developed a deep learning seasonal forecasting system (IceNet) predicting the probability that SIC exceeds 15 %. IceNet significantly outperforms the European Centre for Medium-Range Weather Forecasts (ECMWF) SEAS5 dynamical seasonal prediction system (Johnson et al., 2019) for lead times from 2 to 6 months, and runs over 2000 times faster on a laptop than SEAS5 on a supercomputer. While many studies have investigated such approaches for sea ice forecasting, most of them were not focused on operational short-term forecasting. Grigoryev et al. (2022) developed short-term (up to 10 days) data-driven SIC forecasts for several Arctic seas in an operational context with considering real-time availability of data. Their forecasts, based on U-Net convolutional neural networks (Ronneberger et al., 2015) with predictors from sea ice observations and weather forecasts, significantly outperformed persistence and linear trend forecasts.

Most of the short-term sea ice prediction systems based on machine learning do not use predictors from dynamical sea-ice models (Fritzner et al., 2020; Liu et al., 2021; Grigoryev et al., 2022; Ren et al., 2022; Keller et al., 2023; Kvanum et al., 2024), and it is currently unclear whether adding such predictors would significantly improve forecast accuracy. This study aims at assessing the impact of using predictors from dynamical sea ice models in the development of SIC forecasts from machine learning, as well as the impact of post-processing SIC forecasts from a dynamical sea ice model for lead times from 1 to 10 days. The post-processing method developed is based on convolutional neural networks with a U-Net architecture (Ronneberger et al., 2015), and use predictors from TOPAZ4 SIC forecasts, ECMWF weather forecasts, and SIC observations from the Advanced Microwave Scanning Radiometer 2 (AMSR2). It is evaluated by assessing the improvement compared to the raw TOPAZ4 forecasts, and to predictions from similar deep learning models without using predictors from TOPAZ4 sea ice forecasts. In section 2, the data, the development of the deep learning models, as well as the methods used for evaluating the forecasts are presented. The results are then described in section 3, followed by the discussions and conclusions in section 4.

## 2 Data and methods

### 2.1 Sea ice observations

The AMSR2 sensor is a conically scanning, dual-polarised microwave radiometer that measures the microwave emissions emitted from the Earth's surface across several frequencies. AMSR2 SIC data are currently assimilated into sea ice prediction systems, such as the Barents-2.5km model (Röhrs et al., 2023; Durán Moro et al., 2023), due to its capability of daily coverage of the polar regions and its independence of solar illumination, enabling year-round observation. The AMSR2 SIC observations used in this study were produced using the resolution-enhancing (reSICCI3LF) algorithm, which was initially developed for the European Space Agency Climate Change Initiative (ESA CCI) (Lavergne et al., 2021) and adapted for the AMSR2 mission in the Sea Ice Retrievals and data Assimilation in NOrway (SIRANO) project (Rusin et al., 2024). This algorithm aims at producing high-resolution SIC fields with low measurement uncertainties by combining two retrievals. The 19 and 37 GHz channels are used to derive a coarse SIC field (15 km) with low measurement uncertainties, whereas the 89 GHz channels are used to derive a higher resolution SIC field ($\sim$5km) with larger uncertainties. The high resolution details derived from the 89 GHz channels are then added to the coarse SIC field, enabling the production of a SIC field with low measurement uncertainties at a higher spatial resolution ($\sim$5km). Using this algorithm, daily averaged pan-Arctic SIC fields were produced for the period 2012-2022 on a 5 km Equal-Area Scalable Earth 2.0 (EASE2) grid. In this study, these new observations are used as reference for evaluating the SIC forecasts, as well as for some predictors and the target variable of deep learning models.

In addition, the ice charts produced by the Ice Service of the Norwegian Meteorological Institute (https://www.cryo.met.no/en/latest-ice-charts; JCOMM Expert Team on sea ice, 2017) are used as an independent dataset for evaluating the AMSR2 SIC observations and the forecasts developed in this study. The ice charts are manually drawn by ice analysts using several types of remote sensing data. Due to their high spatial resolution, synthetic-aperture radar (SAR) images constitute the main source of information where they are available. Elsewhere, visible and infrared observations are used in priority, while passive microwave retrievals are used where no other observations are available. For evaluating the SIC forecasts, the ice charts were interpolated on the grid used for the deep learning models using nearest neighbor interpolation. It is worth noting that the ice charts provide SIC categories and are not produced during weekends. Therefore, the number of ice charts available in 2022 for evaluating the SIC forecasts varies depending on lead time (between 144 and 243), and is considerably lower than the number of AMSR2 SIC observations available.

### 2.2 Predictors and data sets used for the deep learning models

The post-processing method developed in this study is applied to TOPAZ4 sea ice forecasts. TOPAZ4 is a numerical prediction system producing 10-day forecasts at 12.5 km resolution for the Arctic and the North Atlantic with hourly time steps (Sakov et al., 2012). It consists of a sea ice model with one thickness category and an elastic–viscous–plastic rheology (Hunke and Dukowicz, 1997) coupled with the version 2.2 of the Hybrid Coordinate Ocean Model (HYCOM; Bleck, 2002; Chassignet et al., 2006). Sea ice and oceanic observations are assimilated weekly using an ensemble Kalman filter, and the ocean surface is forced by ECMWF high resolution weather forecasts.

**Table 1.** List of predictors used for the deep learning models.

| Source | Variable | Time |
|---|---|---|
| AMSR2 | SIC observations | Day preceding the forecast start date |
| AMSR2 | SIC trend | 5 days preceding the forecast start date |
| ECMWF | 2-meter temperature | Mean value between the forecast start date and the predicted lead time |
| ECMWF | 10-meter x wind component | Mean value between the forecast start date and the predicted lead time |
| ECMWF | 10-meter y wind component | Mean value between the forecast start date and the predicted lead time |
| TOPAZ4 | Land sea mask | Constant predictor |
| TOPAZ4 | SIC forecasts | Predicted lead time |
| TOPAZ4 and AMSR2 | TOPAZ4 initial errors | Day preceding the forecast start date and 1-day lead time |

Wind and temperature high-resolution forecasts (HRES) from the ECMWF Integrated Forecasting System (IFS) are also used as predictors. These forecasts have lead times up to 10 days and are produced 4 times per day, but only the forecasts starting at 00:00 UTC are used in this study. Due to the developments of IFS HRES over time, forecasts produced by different model cycles have been used, and it is worth noting that the spatial resolution has changed from about 16 to 9 km in March
2016 (https://www.ecmwf.int/en/forecasts/documentation-and-support/changes-ecmwf-model).

In this work, the deep learning models have been developed using 8 predictors that can be divided into three categories (table 1). First, two predictors are derived from AMSR2 SIC observations acquired before the forecast start date, and consist of the SIC observations from the day preceding the forecast start date, and the SIC trend calculated over the 5 days preceding the forecast start date (in % per day). The second category consists of weather forecasts from ECMWF that have been averaged
between the forecast start date and the predicted lead time. These predictors are the 2-m temperature, as well as the x and y components of the 10-m wind on the grid used for the deep learning models. Then, predictors from the TOPAZ4 ocean model can be considered as the last category. These variables are the SIC forecasts for the predicted lead time, the difference between TOPAZ4 SIC during the first daily time step and the SIC observed the day before (hereafter referred to as "TOPAZ4 initial errors"), and the land sea mask (constant predictor).

The predictors from weather and sea ice forecasts vary depending on lead time. Therefore, different deep learning regression models were developed for each lead time from 1 to 10 days. Before developing the deep learning models, all the predictors and the SIC observations used for the target variable were projected onto a common grid using nearest neighbor interpolation. This grid has the same projection and spatial resolution (12.5 km) as the TOPAZ4 prediction system, but is smaller (544 x 544) due to the constraints related to the U-Net architecture (the x and y axes must be divided by 2 several times). Nevertheless,
this grid includes all the grid points that can potentially be covered by sea ice from the TOPAZ4 prediction system. When providing the predictors to the neural networks, all the grid points must contain valid values, meaning that the land grid points must be filled with valid values for oceanic variables. In this study, the land grid points were considered as ice-free ocean in the predictors. Furthermore, all the predictors and the target variable have been normalized (resulting in values ranging from

0 to 1) before providing them to the neural networks. The training data set was used to compute the minimum and maximum values of the variables, which were then used for the normalization.

Though TOPAZ4 produces 10-day forecasts daily, only the forecasts starting on Thursdays (when data assimilation is performed) are stored in the long-term archive. Therefore, weekly data during the period 2013 - 2020 were used for training the deep learning models, resulting in about 400 forecasts for each lead time. However, we stored daily TOPAZ4 forecasts from 2021, and we therefore used daily data for the validation and test data sets, which consist of the forecasts from 2021 and 2022, respectively.

## 2.3 Development of the deep learning models

U-Net neural networks are designed to perform image segmentation tasks using an encoder-decoder architecture (Ronneberger et al., 2015), and have been successfully used in earlier studies for sea ice forecasting (Andersson et al., 2021; Grigoryev et al., 2022; Keller et al., 2023; Kvanum et al., 2024). Several variations from the original U-Net architecture of Ronneberger et al. (2015) are tested in our study. First, some models were developed using residual connections (He et al., 2016) in the convolutional blocks (meaning that the residual was learned at each block), which was shown to ease neural network training (He et al., 2016). It is worth noting that the residual U-Net architecture was used by Keller et al. (2023) for predicting the sea ice extent in the Beaufort sea. Furthermore, the impact of using attention blocks introduced by Oktay et al. (2018) in the decoder, and designed to give more weight (attention) on areas that are challenging to predict (these regions are identified by the attention blocks during training), is also evaluated. The benefit of using attention blocks for sea ice forecasting was already shown by Ren et al. (2022) who developed an attention block (different from the one used in this study) for sea ice prediction with a fully convolutional network. Finally, average pooling was used in the downsampling blocks of the encoder instead of max pooling due to slightly better performances observed during the tuning phase (see supplement).

In the original U-Net architecture (Ronneberger et al., 2015), the number of convolutional filters is doubled (divided by two) at every layer in the encoder (decoder). We used the same strategy with 32 convolutional filters in the first layer, and with the He weight initialization technique (He et al., 2015). 5 downsampling and 5 upsampling operations were used in the neural networks, resulting in feature maps with a size of 17 x 17 grid points in the bottleneck (compared to 544 x 544 grid points in the predictors). The models were trained using 100 epochs and a batch size of 4. An Adam optimizer was used with an initial learning rate of 0.005, which was then divided by 2 every 25 epochs. The mean squared error was used as loss function and the model version with the best validation loss was selected during training in order to avoid overfitting. Training the models, which contain between 31 and 39 million parameters, takes about 3 hours on a 12 GB GPU (NVIDIA Tesla P100 PCIe). For further details regarding the model architectures, note that the codes used for creating the deep learning models are publicly available in a GitHub directory (see code availability section).

## 2.4 Verification scores

The forecasts are evaluated using two verification scores in this study. In order to analyze the full range of SIC values in the forecasts, as well as to strongly penalize large errors, the root mean square error (RMSE) is calculated over all oceanic grid

points. In addition, the sea ice edge position is also evaluated. While the ice edge is defined here by the 15 % SIC contour (excluding coastlines) when the AMSR2 SIC observations are used as reference, the 10 % SIC contour is used when the forecasts are compared to the ice charts from the Norwegian Meteorological Institute (the 10 % SIC contour separates two sea ice categories). The Integrated Ice Edge Error (IIEE; Goessling et al., 2016) divided by the observed ice edge length (hereafter referred to as "ice edge distance error") is used for evaluating the ice edge positions, and the ice edge length is assessed using the method introduced by Melsom et al. (2019). While the IIEE measures the area of mismatch between two data sets, the ice edge distance error (Melsom et al., 2019) assesses the mean distance between two ice edges. The ice edge distance error has also the advantage of being less seasonally dependent than the IIEE which is greatly influenced by the ice edge length (Goessling et al., 2016; Palerme et al., 2019). Therefore, it is more suitable than the IIEE for comparing and averaging forecast scores from different seasons. Furthermore, the Wilcoxon signed-rank test is used in this study to analyze the statistical significance of the differences between the forecast scores due to its relevance for paired observations (the same observations are used for evaluating different forecasts) and for non-parametric data (the errors are not normally distributed for SIC). This analysis was performed using the two-tailed hypothesis with a significance level of 0.05. It is worth noting that the Wilcoxon signed-rank test assesses the statistical significance between the differences in the distribution of the errors (and not between the mean errors).

## 2.5 Benchmark forecasts

The performances of the deep learning models are evaluated by assessing the improvement compared to the raw TOPAZ4 forecasts. In addition, several benchmark forecasts are used as reference. First, persistence of the AMSR2 SIC observations from the day preceding the forecast start date (hereafter referred to as "persistence of AMSR2 SIC") is used, and can be considered as the limit from which the forecasts are skillful. When the forecasts are evaluated using the ice charts as reference, a similar benchmark forecast consisting of persistence of the ice charts from the day preceding the forecast start date is also used (hereafter referred as "persistence of the ice charts"). The second benchmark forecast (hereafter referred to as "anomaly persistence") consists of calculating the SIC anomalies from AMSR2 observations compared to a climatological reference the day before the forecast start date, and adding these initial anomalies to the climatology during the target date. Then, the values lower than 0 % and higher than 100 % are assigned to 0 and 100 %, respectively. All the full years between the launch of AMSR2 (May 2012) and the test period (2022) were used for calculating the climatology, resulting in a 9-year period (2013 - 2021). The last benchmark forecast consists of calculating the difference between TOPAZ4 SIC during the first daily time step and the SIC observed the day before (in order to use only observations available at the forecast start date), and then subtracting this difference from the TOPAZ4 forecasts for each lead time (hereafter referred to as "TOPAZ4 bias corrected"). The resulting values lower than 0 % and higher than 100 % are then assigned to 0 and 100 %, respectively. Note that this forecast is equal to persistence of AMSR2 SIC for 1-day lead time.

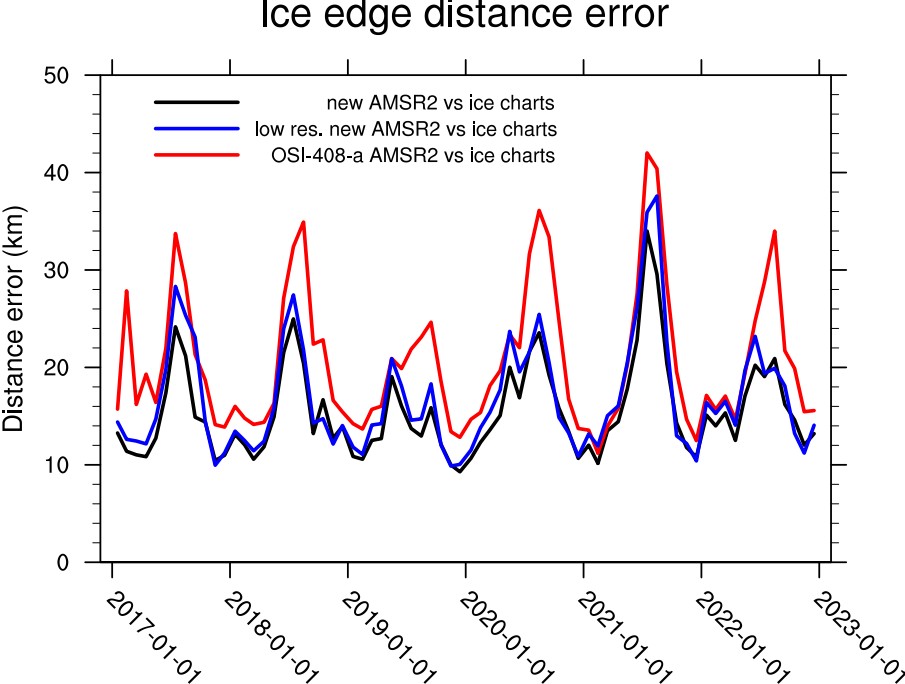

**Figure 1.** Evaluation of the ice edge positions from the new AMSR2 sea ice concentration observations used in this study and the product OSI-408-a from the Ocean and Sea Ice Satellite Application Facility (OSI SAF) during the period 2017-2022. The ice charts produced by the Ice Service of the Norwegian Meteorological Institute are used as reference, and the analysis has therefore been done in the area covered by the ice charts (European Arctic). The ice edge distance error (see section 2.4) is used for calculating the mean distance between the ice edges, and the monthly mean distances are reported in this figure. The red and blue lines correspond to the ice edge distance errors after all products were integrated onto the 10 km OSI-408-a grid. The black line shows the ice edge distance error for the new AMSR2 SIC product on its 5 km grid, thus retaining information on the finer resolution.

## 3 Results

### 3.1 Sea ice concentration observations

The new AMSR2 observations were evaluated and compared to the Ocean and Sea Ice Satellite Application Facility (OSI-SAF) product OSI-408-a, which is also based on AMSR2 retrievals but with a spatial resolution of 10 km. The position of the ice edge (defined by the 10 % SIC contour here) was evaluated during the period from 2017 to 2022 using the ice charts from the Norwegian Meteorological Institute (JCOMM Expert Team on sea ice, 2017) as reference. All the data sets were projected onto the grid of the OSI-408-a product using nearest neighbor interpolation, but only the area covered by the ice charts (European

Arctic) was taken into account for this evaluation. The mean distances between the ice edges from the AMSR2 products and from the ice charts were assessed using the ice edge distance error. Overall, the new AMSR2 data set outperforms the OSI-408-a product (figure 1), with mean values of 16.8 km and 20.6 km for the new AMSR2 observations and the OSI-408-

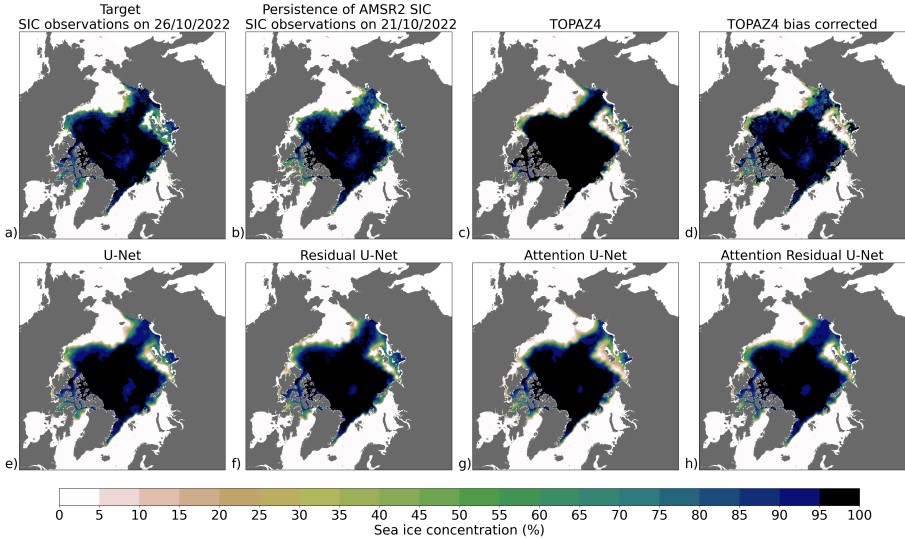

**Figure 2.** 5-day sea ice concentration forecasts from different forecasting systems initialized on 22/10/2022. a) AMSR2 sea ice concentration observations on 26/10/2022 (target date). b) AMSR2 sea ice concentration observations during the day preceding the forecast start date (21/10/2022). 5-day sea ice concentration forecasts from different systems: TOPAZ4 (c), TOPAZ4 bias corrected (d), deep learning model with the U-Net architecture (e), deep learning model with the Residual U-Net architecture (f), deep learning model with the Attention U-Net architecture (g), deep learning model with the Attention Residual U-Net architecture (h).

a product, respectively. Moreover, the new AMSR2 observations particularly outperform the OSI-408a product close to the sea ice minimum (in August, September and October) compared to the rest of the year. In order to assess the impact of the resolution, a supplementary analysis was performed on the 5 km grid from the new AMSR2 SIC observations with interpolating the ice charts onto this grid. On the 5 km grid, the mean distance between the ice edges from the new AMSR2 observations and the ice charts is 15.4 km, adding further confidence in the quality of the new product.

## 3.2 Model architectures

The original U-Net architecture (with average pooling instead of max pooling) is compared to architectures including residual and attention blocks in figures 2 and 3. It is worth noting that the architecture influences the number of model parameters, which can also influence the performances. The number of parameters varies from 31 million for the U-Net models to 39 million for the Attention Residual U-Net models, and the models with the Residual U-Net and Attention U-Net architectures contain about 33 and 37 million parameters, respectively. Figure 2 shows 5-day forecasts initialized on 22/10/2022 from TOPAZ4, TOPAZ4 bias corrected, and deep learning models developed with different architectures. Between the day preceding the forecast start date (21/10/2022) and the target date (26/10/2022), the sea ice cover has increased in the Laptev and East Siberian seas, as well as in the Baffin Bay. Moreover, a few large polynyas were located around New Siberian Islands during the target date, in an area not covered by sea ice during the day preceding the forecast start date. While all the deep learning models, as well as

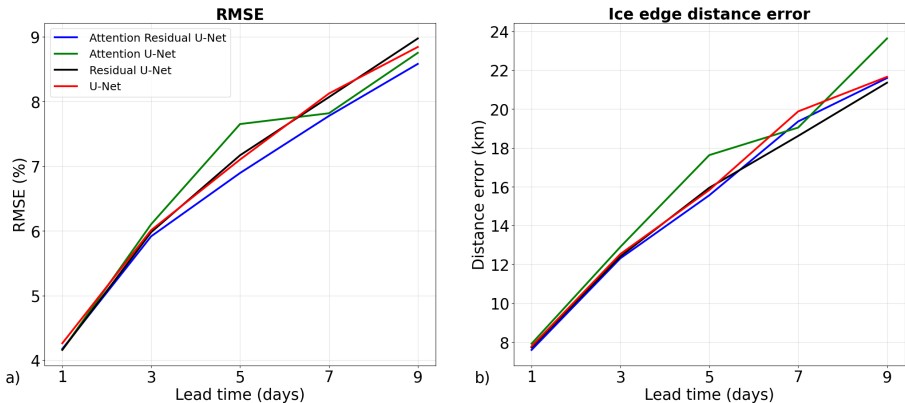

**Figure 3.** Comparison of the performances of deep learning models with different architectures during 2021 (validation period). a) Root mean square error (RMSE) of the sea ice concentration. b) Mean error for the sea ice edge position defined by the 15 % sea ice concentration contour (ice edge distance error). AMSR2 sea ice concentration observations are used as reference.

TOPAZ4 and TOPAZ4 bias corrected, reproduce an increase in sea ice cover in the Laptev and East Siberian seas, only the deep learning models predicted an increase in sea ice cover in the Baffin Bay. The model with the Attention U-Net architecture produces very small positive SIC (often lower than 2 %) in large areas where no sea ice is observed during the target date, which is a pattern often observed with this model for other dates as well. Nevertheless, it seems that adding residual blocks to this model (resulting in the Attention Residual U-Net architecture) consistently helps to better predict these areas. Furthermore, the model with the Attention Residual U-Net architecture produces the most realistic forecasts of the polynyas among the deep learning models.

In figure 3, the performances of the deep models with different architectures are evaluated during the validation period (2021). For 1-day lead time, the different architectures produce forecasts with similar performances, except the U-Net architecture for which the forecasts have a RMSE about 2 % larger. The models with the Attention Residual U-Net architecture have the lowest RMSE for longer lead times, and the lowest errors for the position of the ice edge for lead times up to 5 days. Therefore, the Attention Residual U-Net architecture has been selected for the rest of this study despite the higher errors for the position of the ice edge for 7 and 9-day lead times compared to the forecasts produced using the Residual U-Net architecture. Furthermore, it is worth noting that the forecasts produced using the Attention Residual U-Net architecture have lower RMSE and lower errors for the position of the ice edge than the forecasts from the models with the U-Net architecture for all lead times. These differences are statistically significant (p-value from the Wilcoxon signed-rank test < 0.05) for all lead times and metrics, except for the ice edge distance error for 9-day lead time.

## 3.3 Performances of the deep learning models

In figure 4, the predictions from the models with the Attention Residual U-Net architecture are compared to the benchmark forecasts during the test period (2022) using AMSR2 SIC observations as reference. They significantly outperform all the

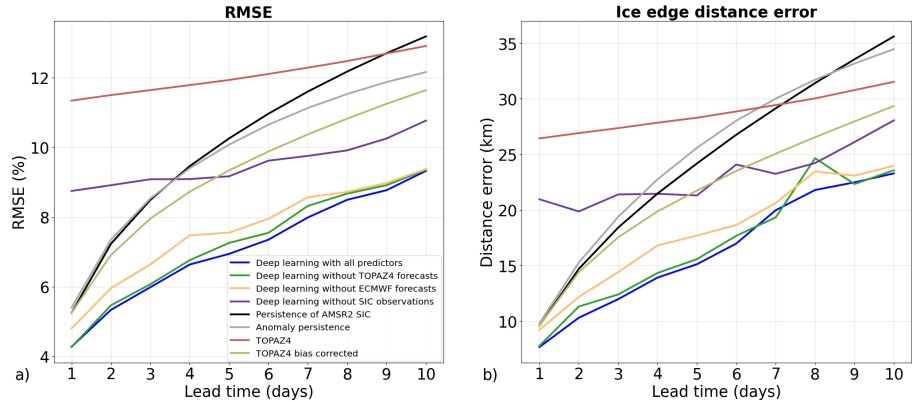

**Figure 4.** Performances of the deep learning models with the Attention Residual U-Net architecture during 2022 (test period) using the AMSR2 sea ice concentration observations as reference. The deep learning models using all predictors are shown by the blue curves, the models which do not use predictors from TOPAZ4 sea ice forecasts (sea ice concentration forecasts and initial errors) are shown by the green curves, the models which do not use predictors from ECMWF weather forecasts (2-m temperature and wind) are shown by the yellow curves, and the models which do not use predictors from sea ice observations (AMSR2 sea ice concentration, AMSR2 sea ice concentration trend, and TOPAZ4 initial errors) are shown by the purple curves.

benchmark forecasts for all lead times. The RMSE is improved on average by 41 % compared to TOPAZ4 (between 28 % and 62 % depending on lead times), by 29 % compared to persistence of AMSR2 SIC (between 19 % and 33%), by 23 % compared to TOPAZ4 bias corrected (between 19 % and 26 %), and by 27 % compared to anomaly persistence (between 21 % and 31 %). Furthermore, the ice edge distance error is reduced on average by 44 % compared to TOPAZ4, by 25 % compared to TOPAZ4 bias corrected, by 32 % compared to persistence of AMSR2 SIC, and by 34 % compared to anomaly persistence.

In order to assess the impact of the different data sets used in the predictors (observations, sea ice and weather forecasts), other deep learning models were developed without including either predictors from TOPAZ4 sea ice forecasts (SIC forecasts and TOPAZ4 initial errors), predictors from ECMWF weather forecasts (temperature and wind forecasts), or predictors from AMSR2 SIC observations (SIC during the day preceding the forecast start date, SIC trend, and TOPAZ4 initial errors). These models have the same architecture and hyperparameters as the models using all predictors, and their performances are also shown in figure 4. Note that TOPAZ4 initial errors is considered as a predictor from TOPAZ4 sea ice forecasts and from AMSR2 SIC observations in this experiment since both data sets are needed to create this predictor. Overall, the predictions are much more impacted by dropping ECMWF weather forecasts than by removing TOPAZ4 sea ice forecasts. On average, the relative increase in RMSE is 2.1 % if the predictors from TOPAZ4 sea ice forecasts are removed compared to 7.7 % if the predictors from ECMWF weather forecasts are removed. The differences in RMSE between the models using all predictors and those developed without ECMWF weather forecasts are statistically significant for all lead times (p-value from the Wilcoxon signed-rank test < 0.05). When comparing the models using all predictors to those developed without TOPAZ4 sea ice forecasts, the differences in RMSE are statistically significant for all lead times, except 1 and 10 days. Furthermore, the forecasts from

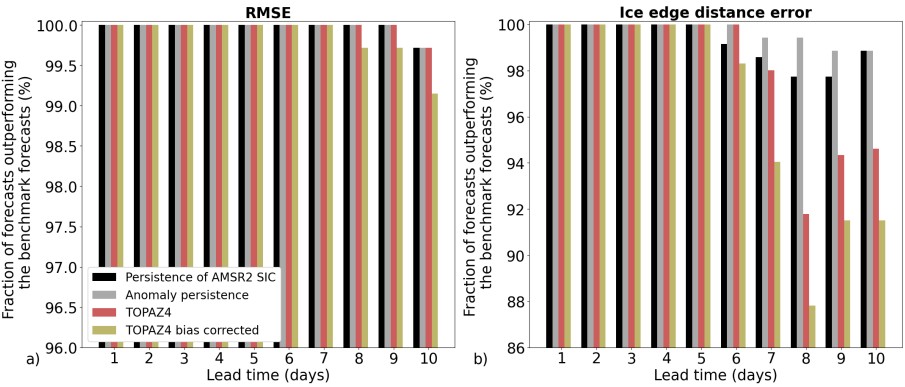

**Figure 5.** Fraction of days in 2022 (test period) during which the forecasts from the models with the Attention Residual U-Net architecture outperform the different benchmark forecasts when the forecasts are evaluated with the RMSE (a) and with the ice edge distance error (b). AMSR2 sea ice concentration observations are used as reference.

ECMWF and TOPAZ4 have relatively similar impacts on the RMSE for lead times from 8 to 10 days. The differences in RMSE between the models developed without TOPAZ4 sea ice forecasts and those developed without ECMWF weather forecasts remain statistically significant for lead times up to 9 days, but this difference is not significant for 10-day lead time.

The impact of removing predictors from TOPAZ4 or ECMWF forecasts is stronger for the position of the ice edge, with a mean increase in ice edge distance error of 3.5 % and 12.3 % for the predictors from TOPAZ4 and ECMWF forecasts, respectively. Nevertheless, the models developed without TOPAZ4 sea ice forecasts have slightly smaller ice edge distance errors than the models using all predictors for lead times of 7 and 9 days, and the difference in ice edge distance error is not statistically significant for 10-day lead time. Furthermore, removing the predictors from sea ice observations has a very strong impact on the predictions, with a mean relative increase of 39 % in RMSE and of 55 % in ice edge distance error.

Figure 5 shows the fraction of days in 2022 during which the forecasts produced by the deep learning models outperform the different benchmark forecasts. When the forecasts are evaluated using the RMSE, the forecasts from the deep learning models outperform all benchmark forecasts for lead times from 1 to 7 days, and at least 99 % of the different benchmark forecasts for longer lead times. Moreover, the forecasts from the deep learning models outperform all benchmark forecasts for lead times from 1 to 5 days when the ice edge position is evaluated. For longer lead times, the deep learning models outperform at least 97 % of persistence of AMSR2 SIC forecasts and 98 % of the anomaly persistence forecasts. They also predict the ice edge position with better accuracy than TOPAZ4 in at least 91 % of the cases for all lead times, and in at least 87 % of the cases compared to TOPAZ4 bias corrected.

In order to assess the performances of the SIC forecasts using independent observations, an additional evaluation was performed in the European Arctic using the ice charts from the Norwegian Meteorological Institute as reference (figure 6). Since the ice charts provide sea ice categories (and not SIC as a continuous variable), only the ice edge position is evaluated in figure 6. On average, the forecasts from the deep learning models have an ice edge distance error 40 % lower than TOPAZ4 forecasts, 23 % lower than TOPAZ4 bias corrected, 29 % lower than persistence of AMSR2 SIC, and 22 % lower than persistence of the

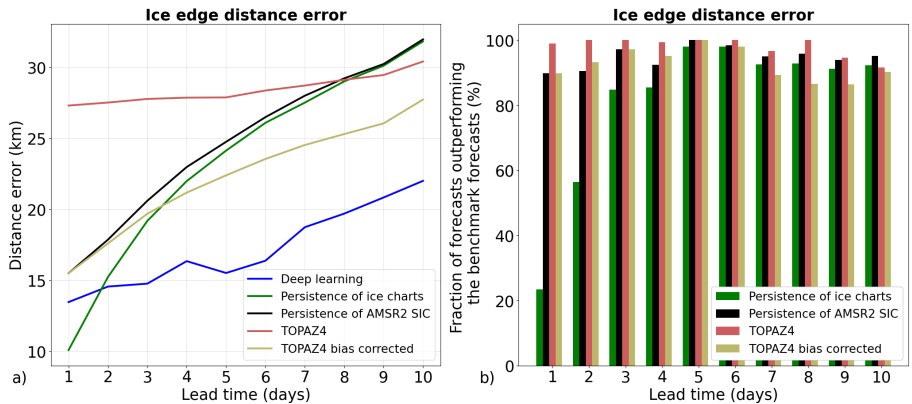

**Figure 6.** Performances of the deep learning models with the Attention Residual U-Net architecture during 2022 (test period) using the ice charts as reference. The ice edge position (defined by the 10 % SIC contour) is evaluated. a) Mean ice edge distance errors depending on lead time. b) Fraction of days in 2022 during which the forecasts from the models with the Attention Residual U-Net architecture outperform the different benchmark forecasts when the forecasts are evaluated using the ice edge distance error. It is worth noting that this evaluation is performed over the area covered by the ice charts from the Norwegian Meteorological Institute (European Arctic), and that the number of forecasts evaluated varies depending on lead time because ice charts are not produced during weekends.

ice charts. While the forecasts from the deep learning models outperform TOPAZ4, TOPAZ4 bias corrected, and persistence of AMSR2 SIC for all lead times, they have worse performances than persistence of the ice charts for 1-day lead time (the ice edge distance error is 33 % larger). Moreover, only 23 % of the forecasts from the deep learning models outperform persistence of the ice charts for 1-day lead time. Nevertheless, the forecasts from the deep learning models significantly outperform persistence of the ice charts for longer lead times (p-value from the Wilcoxon signed-rank test < 0.05).

### 3.4 Predictor importances

In order to analyze the impact of each predictor on the forecasts, two approaches are used in this study. The first method is the same as the one used in figure 4 to test the impact of removing some data sets from the list of predictors, except that only one predictor is removed for each model. Then, the performances of the different models are compared to assess the impact of the different predictors on the forecasts. Due to the relatively long computing time necessary for developing the different models, this experiment has only been performed using half of the lead times. While two predictors are used for the wind forecasts (x and y components), only one model per lead time was developed with removing both predictors simultaneously to test the impact of wind forecasts. It is worth noting that the importance of highly correlated predictors can be underestimated using this method since similar information is provided to the neural network when one predictor is removed. The results from this experiment are shown in figure 7. While all the predictors tend to reduce the RMSE averaged over all lead time, some predictors have a negative impact on the predictions of the ice edge (ECMWF 2-m temperature forecasts, AMSR2 sea ice concentration observations and trend). The wind forecasts have the largest impact among the predictors for all lead times. Removing the wind

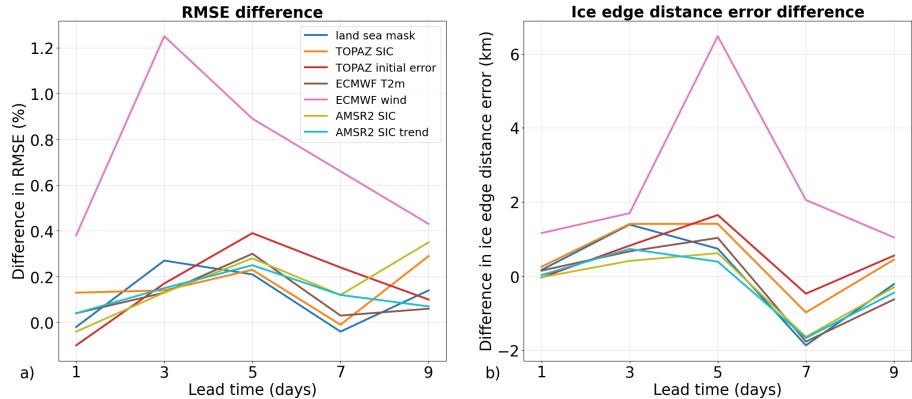

**Figure 7.** Differences in root mean square error of the sea ice concentration (a) and in ice edge distance error (b) when one of the predictor variables is not used in the deep learning models during 2022 (test period). The differences represent the subtraction between the performances of the models in which one predictor was not used and the models using all the predictors. Therefore a positive value means that adding the variable in the model improves the forecasts. AMSR2 sea ice concentration observations are used as reference.

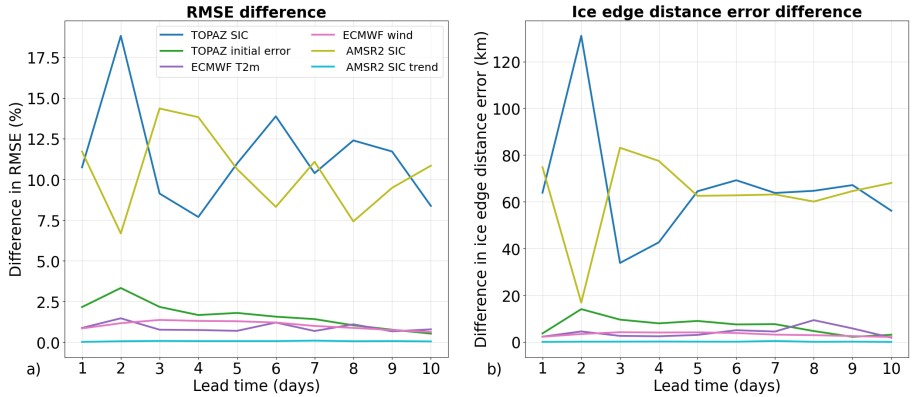

**Figure 8.** Differences in root mean square error of the sea ice concentration (a) and in ice edge distance error (b) when the field from a wrong date is provided to the deep learning models for one predictor during 2022 (test period). The differences represent the subtraction between the performances of the models in which one predictor is shuffled and the reference model. AMSR2 sea ice concentration observations are used as reference.

forecasts leads to a mean absolute increase in RMSE of 0.72 % and a mean increase in ice edge distance error of 2.49 km. The other predictors have a much lower impact on the forecasts. Overall, the predictors from TOPAZ4 (SIC forecasts and initial errors) have the strongest impact on the predictions of the ice edge among the other predictors, with a mean difference in ice edge distance error of about 0.5 km for each predictor. However, the predictors from TOPAZ4 sea ice forecasts have a slight negative impact on the 7-day forecasts of the ice edge position.

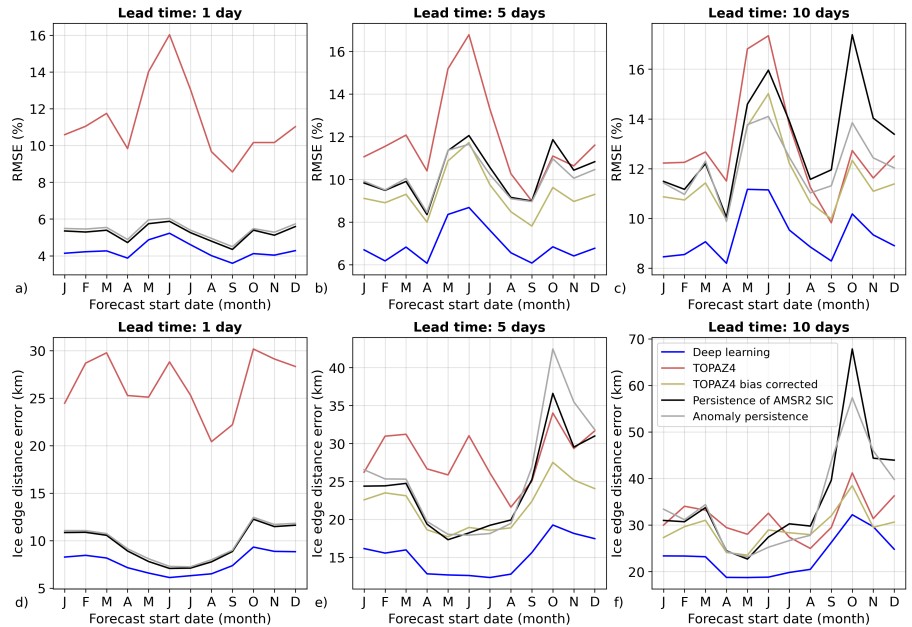

**Figure 9.** Seasonal variability in the performances of the deep learning models with the Attention Residual U-Net architecture in 2022 (test period) for different lead times (1, 5, and 10 days) when the forecasts are evaluated using the RMSE (a, b, c) and using the ice edge distance error (d, e, f). AMSR2 sea ice concentration observations are used as reference.

Another method called permutation feature importance has been used to assess the impact of the different predictors on the forecasts (figure 8). In this method, only the models developed using all predictors are used. When making a forecast, one predictor is randomly permuted by providing the predictor data from another forecast start date. The goal of this experiment is to test how much the models are fitted to the different predictors. Figure 8 shows that the neural networks are considerably fitted on the TOPAZ4 SIC forecasts and the AMSR2 SIC observations. Permuting the fields from these predictors produces very inaccurate forecasts, leading to mean absolute increases in RMSE of 11.4 % and 10.4 % if the TOPAZ4 SIC forecasts and the AMSR2 SIC observations are permuted, respectively. Similar results were obtained for the position of the ice edge, with large increases in the ice edge distance error if these predictors are permuted (65.7 km and 63.3 km for TOPAZ4 SIC forecasts and AMSR2 SIC observations, respectively). Moreover, the relative importances of these two predictors seem anti-correlated depending on lead times. This suggests that the neural networks need at least one SIC field to guide the SIC predictions. Furthermore, permuting the AMSR2 SIC trend seems to have almost no impact on the forecasts, suggesting that the neural networks use this predictor only marginally.

### 3.5 Seasonal and spatial variabilities

Figure 9 shows the seasonal variability in the performances of the deep learning models for lead times of 1, 5, and 10 days. Overall, the deep learning models show robust results, with no clear seasonal cycle in the relative improvement compared

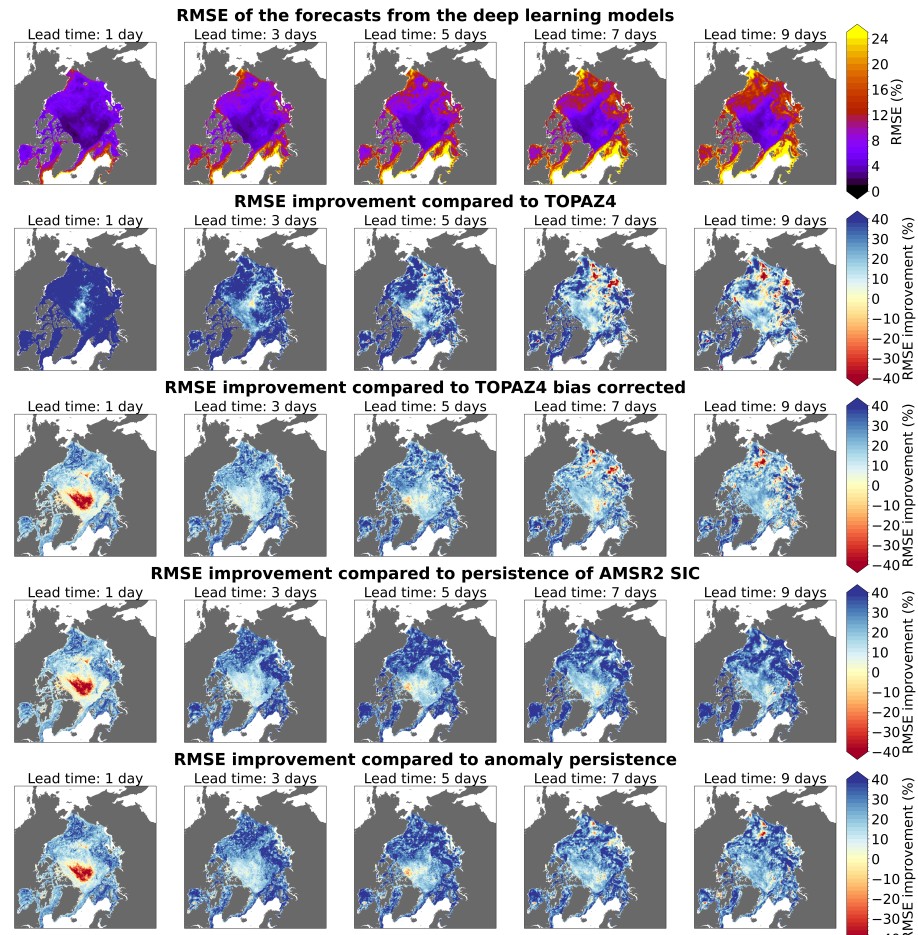

**Figure 10.** Root mean square error (RMSE) of the forecasts from the deep learning models with the Attention Residual U-Net architecture (first row) in 2022 (test period). Relative improvement in RMSE (%) compared to TOPAZ4 forecasts (second row), TOPAZ4 bias corrected (third row), persistence of AMSR2 SIC (fourth row), and anomaly persistence (fifth row). Positive values mean that the deep learning forecasts outperform the benchmark forecasts. AMSR2 sea ice concentration observations are used as reference, and the grid points with less than 50 days during which the AMSR2 observations indicate some sea ice (sea ice concentration higher than 0 %) are not taken into account in this figure.

to TOPAZ4 forecasts and persistence of AMSR2 SIC. Moreover, the deep learning models outperform all the benchmark forecasts for all the months, except in November when the 10-day forecasts are evaluated using the ice edge distance error. In November, the 10-day forecasts from the deep learning models have similar ice edge distance error as the TOPAZ4 bias corrected forecasts.

The spatial variability in the performances of the deep learning models in 2022 is shown in figure 10. The grid points with less than 50 days during which the AMSR2 observations indicate some sea ice (SIC higher than 0 %) are excluded from the

analysis in order to keep only meaningful data. Nevertheless, figure 10 must be interpreted carefully because forecasts from different seasons with varying sea ice edge positions are taken into account in this analysis. The forecasts from the deep learning models outperform the TOPAZ4 forecasts almost everywhere, but have slightly lower performances in the East Siberian sea compared to the rest of the Arctic. Nevertheless, it is difficult to determine if these poorer performances in the East Siberian sea are persistent because only one year is used for this analysis. Furthermore, the relative improvement from the forecasts produced by the deep learning models compared to TOPAZ4 forecasts decreases with increasing lead times. Compared to persistence of AMSR2 SIC and anomaly persistence, the relative improvement in RMSE increases with increasing lead times. There is an area in the Central Arctic where the 1-day forecasts from the deep learning models have larger RMSE than TOPAZ4 bias corrected, persistence of AMSR2 SIC, and anomaly persistence. However, the forecasts from the deep learning models have low RMSE in this area, meaning that the relative differences in this area do not represent large absolute values. Except for this area in the Central Arctic for 1-day lead time, the forecasts from the deep learning models outperform the benchmark forecasts almost everywhere, with larger improvements in areas where the marginal ice zone is often located.

## 4 Discussion and conclusion

The forecasts from the deep learning models developed in this study significantly outperform all the benchmark forecasts for all lead times when the AMSR2 SIC observations are used as reference, with a mean RMSE 41 % lower than for TOPAZ4 forecasts and 29 % lower than for persistence of AMSR2 SIC. They also considerably better predict the ice edge position than the benchmark forecasts (the ice edge distance error is reduced by 44 % and 32 % compared to TOPAZ4 and persistence of AMSR2 SIC, respectively). Moreover, their good performances for various seasons and locations, as well as the relatively similar results obtained during the validation and test periods (see supplement), suggest that these models are robust. While it takes less than a second to predict the sea ice concentration for one lead time on a 12 GB GPU (NVIDIA Tesla P100 PCIe) once the list of predictors is available, the full processing chain including the production of the predictors on a common grid takes about 4 minutes for all lead times. This is negligible compared to the time necessary for producing TOPAZ4 forecasts, and therefore reasonable in an operational context. However, the production of TOPAZ4 forecasts was stopped in April 2024, and the AMSR2 SIC observations used in this study are not available in near real time yet. This prevents the operational use of the post-processing method presented here.

Using the ice charts from the Norwegian Meteorological Institute as reference, the forecasts from the deep learning models outperform all benchmark forecasts for lead times longer than 1 day in the European Arctic, but are worse than persistence of the ice charts for 1-day lead time. Since the deep learning models are trained using AMSR2 SIC observations for the target variable, it cannot be expected that they perform better than the differences between the two observational products (figure 1). While using ice charts for training deep learning models has been recently proposed by Kvanum et al. (2024), this does not allow to predict the SIC as a continuous variable.

Whereas previous studies used the original U-Net architectures for SIC predictions (Andersson et al., 2021; Grigoryev et al., 2022), our results suggest that slightly better performances can be achieved by adding residual and attention blocks

(RMSE about 2.8 % lower on average), resulting in the Attention Residual U-Net architecture. In addition to the original U-Net architecture, Grigoryev et al. (2022) also tested a recurrent U-Net architecture in order to take into account the temporal evolution of the sea ice before the forecast start date. They obtained slightly better results with the recurrent U-Net architecture for short lead-times (until 5 days in the Labrador and Laptev seas and until 10 days in the Barents sea), but worse than with the original U-Net architecture for longer lead times. Furthermore, they reported that the computational cost for training the recurrent U-Net models was much higher than for training the U-Net models. In this study, training the models with the Attention Residual U-Net architecture took about the same time as training the models with the U-Net architecture, and the models with the Attention Residual U-Net architecture have better performances than the models with the U-Net architecture for all lead times.

Including predictors from ECMWF weather forecasts (particularly the wind) has a considerable impact on the SIC predictions, resulting in a 7.7 % reduction in RMSE. This is consistent with the findings from Grigoryev et al. (2022) who assessed the impact of using predictors from weather forecasts produced by the National Centers for Environmental Prediction (NCEP) Global Forecast System (GFS), and reported significant improvements when these predictors are included in their U-Net models. Nevertheless, the impact of ECMWF weather forecasts decreases with increasing lead times in our study. This could be due to the lower skill of weather forecasts for longer lead times, as well as to the pre-processing of these variables before providing them to the neural networks. Averaging the weather forecasts between the forecast start date and the predicted lead time could decrease the impact of these predictors for long lead times. This could be mitigated by providing several predictors covering different lead time ranges to the neural networks, but with the disadvantage of increasing the computational cost.

The impact of using predictors from TOPAZ4 sea ice forecasts is much lower since these predictors lead to a reduction in RMSE of only 2.1 % on average. While the impact of using sea ice forecasts from TOPAZ4 is limited in this study, this does not mean that using predictors from sea ice forecasts does not have stronger potential. TOPAZ4 is an operational system that has been constantly developed since 2012, which can lead to inconsistencies limiting the impact of these predictors. The production of consistent re-forecasts with operational systems could increase the impact of sea ice forecasts in the development of such methods, and should be recommended in the sea ice community. Furthermore, it is likely that more accurate physical-based sea ice forecasts would have larger potential as predictors for machine learning models.

While this study focused on developing pan-Arctic SIC forecasts at the same resolution as the TOPAZ4 prediction system (12.5 km), there is also a need for higher resolution (kilometer scale) sea ice forecasts (Wagner et al., 2020). This can be addressed by developing regional high resolution prediction systems using deep learning such as the recent works from Keller et al. (2023) and Kvanum et al. (2024). Most studies on sea ice forecasting using machine learning have focused on predicting the SIC and the sea ice edge (e.g. Kim et al., 2020; Fritzner et al., 2020; Liu et al., 2021; Andersson et al., 2021; Grigoryev et al., 2022; Ren et al., 2022), probably due to the larger number of reliable SIC observations available compared to other variables such as thickness, drift, and type. However, predictions of other sea ice variables such as thickness and drift are necessary for seafarers, and additional efforts should be made to better predict these variables as well. Finally, probabilistic forecasts can also be developed using supervised machine learning (Haynes et al., 2023), which should have strong potential for sea ice forecasting at short time scales and would be highly relevant for end-users (Wagner et al., 2020).

*Code availability.* The codes used for this analysis are available in the following GitHub directory (Palerme, 2024):

https://github.com/cyrilpalerme/Calibration_of_short_term_SIC_forecasts/

*Data availability.* The AMSR2 sea ice concentration observations are available on the thredds server of the Norwegian Meteorological Institute (https://thredds.met.no/thredds/catalog/cosi/AMSR2_SIC/catalog.html, v1, April2023), the TOPAZ4 forecasts are distributed by the Copernicus Marine Service (https://data.marine.copernicus.eu/products), and a license is needed to download the operational forecasts from the European Centre for Medium-Range Weather Forecasts (ECMWF).

*Author contributions.* C.P.: conceptualization, analysis (machine learning), writing (original draft), and funding acquisition. T.L.: conceptualization, analysis (remote sensing), writing, and funding acquisition. J.R.: analysis (remote sensing) and writing. A.M.: analysis (verification of satellite observations), writing, and funding acquisition. J.B.: conceptualization and writing. A.F.K.: conceptualization and writing. A.M.S.: production of satellite observations. L.B.: writing and funding acquisition. M.M.: writing and funding acquisition.

*Competing interests.* The authors declare that they have no conflict of interest.

*Acknowledgements.* This work has been carried out as part of the SEAFARING project supported by the Norwegian Space Agency and the Copernicus Marine Service COSI project. Copernicus Marine Service is implemented by Mercator Ocean in the framework of a delegation agreement with the European Union. The new AMSR2 SIC observations were developed with support from the SIRANO project (Norwegian Research Council of Norway, grant No. 302917). The authors would like to thank Jean Rabault Førland for grateful discussions, and Sreenivas Bhattiprolu for sharing the codes of some deep learning models (https://github.com/bnsreenu/python_for_microscopists/). Finally, we thank
the two reviewers for their comments which helped us to improve the manuscript.

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
