# Peer review of "Improving short-term sea ice concentration forecasts using deep learning"

_EGUsphere, 2023_

## Referee Comment (RC1)

**Review of paper "Calibration of short-term sea-ice concentration forecasts using deep learning"**

**1 Content**

The authors Palerme, Lavergne, Rusin, Melsom, Brajard, Kvanum, Sørensen, Bertino and Müller present a study on sea-ice concentration forecasts with lead times up to 10 days. They develop post-processing methods based on supervised machine learning and show that these outperform both TOPAZ4 and persistence sea-ice concentration forecasts. Also, they evaluate the impact of different types of predictors.

**2 General comments**

I thank the authors for submitting this already very mature manuscript, which I enjoyed reading and mostly had no problems comprehending. I got a bit lost when it came to the details of the model architecture, but that may be clearer to someone with a stronger AI background. If your target group is people with little AI background, I would suggest boiling down the technical description and spending a bit more time on explaining the terms (see detailed comments below). One aspect where I see room for improvement is the clarity of the motivation. Support of maritime operations is mentioned very prominently in both Abstract and Introduction. Yet, I find little information on crucial aspects for this like timeliness of the forecasts, infrastructure for providing the forecasts to ships operationally or the spatial resolution of the model outcome, compared to what is needed for navigational needs. So, I was wondering whether the current study is intended to be a proof of concept, with the applicability being the focus of future work. It would be good if the authors could clarify this, and elaborate upon for which other aspects their study is relevant besides supporting of maritime operations.

In summary, I suggest to accept this paper for publication after **minor revisions**.

**3 Specific comments**

**Abstract**

General impression: A very concise and yet comprehensive description of your paper. I find little to criticise.

L1: If operational support is your main motivation, I would suggest to add some brief information on the timeliness of your forecasts to the Abstract, as this is crucial for operational use. By how much does the post-processing delay the availability of forecasts, and does the accuracy increase achieved by post-processing outweigh this?

**Introduction**

General impression: A very good description of the existing literature on sea-ice forecasting. You could be a bit clearer on what the new aspects are that your study adds to the body of literature. I would also like to see a bit more details on the motivation and relevance of your study. It is certainly relevant, but it could be made clearer for what. Also, you could end the Introduction by formulating your research goals/questions, which can then be picked up in the Discussion and conclusion section.

L16: ... is often limited by their inaccuracies: I agree, but a reference would be good nevertheless.

L17-19: If mentioned in Melsom et al. (2019), consider giving the accuracy for summer instead of a yearly mean, since that is when most of the maritime traffic happens. Large seasonal variability should still be mentioned.

L22: I find it a bit counter-intuitive to mention seasonal time scales when speaking about weather forecasts, which for me would be up to ten days. I would suggest to either simply speak about forecasts without specification on weather forecasts, or to refer only to those studies which focus on up to ten days lead time.

Last paragraph: You could be clearer on what precisely the goal of your study is. Which is the gap that you want to fill? Starting the paragraph with a sentence on what the new thing on your study is, and then elaborating on the why, would be a good transition towards the rest of the paper.

**Data**

General impression: Good to understand. Partly contained elements which I would suggest shifting to the Results or Methods section (see comments below).

L78: Good to see that you did this analysis, but I would move it to the Results section and focus on the data description here.

L95: Would be interesting to see later on if that impacts the forecast performance.

L97–116: Contains many elements where you describe what you do with the data, rather than which data you use. Consider shifting these elements to the Methods section, or merging Data and Methods section.

L97–105: I think it would help to add the categories in the table, and sort the table after categories, in the same order as they appear in the text.

L113: Does this influence the training, in the sense that the model is trained for points which are discarded later anyway, potentially diminishing the forecast quality for other,

"real" points?

**Methods**

General impression: Section 3.1 was quite hard to comprehend due to the use of many technical terms, which I myself am not very familiar with. May be different for people with a strong AI background. Sections 3.2 and 3.3 were much clearer to me and do not need improvement, except for one question which I raise below.

L124–143: I find the description quite technical. Explaining, or, if they are not crucially needed, omitting terms like "residual connections", "convolutional/attention blocks" or "average pooling" would help to understand non-AI experts to follow your methodology.

L146: Does this also contain the "fake ocean points" described in L113, and if yes, does that influence the results?

L167: TOPAZ should be TOPAZ4 (also in other places, occurred several times).

**Results**

General impression: As for the other sections, I find little to criticise. I did have some open questions, but nothing major.

L175: Does the varying number of model parameters influence the results, in the sense that you get more precise results for the Attention Residual U-Net models simply because there is a larger number of model parameters to optimize upon?

L184: Did you also see the spurious SIC which the Attention U-Net shows on other dates? Generally, concerning Fig. 2: It is fine to show only one day's maps as example, but it would be good to know if you also looked at other dates and how representative this date is for the overall development, how the RMSE's etc evolve over time. Since the ice refreezes rapidly at this time and the SIC is likely varying less in mid-winter, it would be interesting to see if you can also see that in the temporal development of the RMSE's and the comparison to your benchmarks.

L200–206: Do you know why the TOPAZ4 RMSE is higher than the persistence benchmarks for all lead times? Wouldn't you expect it to be better? It's not the focus of your paper, so don't spend too much room on this, but it was one of the first things which came to my mind when looking at Fig. 4.

L206ff: It would be interesting to see how removing predictors influences the runtime of the model. You could discuss, for example, whether the advantage gained in runtime outweighs the small benefit of including TOPAZ4 predictors, and if this might be reason enough to drop the TOPAZ4 predictors altogether.

Figure 5: It would be good if the colors of the bars would be consistent with those in Figure 4.

**Discussion and conclusion**

The section provides a good summary of your findings, and puts them nicely into perspective with other studies. You could state your conclusions more clearly. This could for example be done by formulating a clear goal or research question(s) at the end of the Introduction which you can pick up and answer here to provide a nice framework for your paper. Also, I am missing a discussion on the practical applicability of your study. In the Abstract and Introduction, you name increasing marine traffic as one reason why short-term sea-ice forecasting is relevant. It would be interesting to discuss in how far your model is ready to support decision making for marine operations: How fast would your products be available? Do you have the means to transfer them in near-real time? Does the precision of your results meet the requirements of the onboard ship personnel? Or is your paper rather a proof of concept that it is possible to achieve high-quality short-term forecasts using deep learning, and the transfer to near-real time applications is something which could be done in future?

---

## Referee Comment (RC2)

Review of "Calibration of short-term sea ice concentration forecast using deep learning"

Overview:

This study by Cyril Palerme and co-authors develops a U-Net deep learning to post-process sea ice concentration forecasts in the Arctic. The model uses predictors from numerical sea ice forecasts, weather forecasts, and satellite sea ice concentration observations, and their sensitivities are examined. Analysis of forecasts over the independent test period of 2022 indicate the deep learning model can outperform several noteworthy benchmark forecasts.

General comments:

I really enjoyed reading the paper and was encouraged by the results. The daily SIC forecast problem at <10 day lead time is a challenging one, and even state of the art numerical prediction models have a difficult time with it. So, it's encouraging to see how deep learning may be able to help in this regard.

Semi-major comments:

I have two semi-major comments, one of which might not be possible to address, and the other might not be a problem at all and just my ignorance.

The first has to do with the verifying observation choice of the "new" AMSR2 product. I appreciate the analysis done in section 2.1 that compares this product against Norwegian ice charts, and I don't doubt that this is a fine product to use for the kind of the high resolution forecasts being produced. However, at such short lead times, I worry that there is an independence problem between one of the most important predictors in the U-Net model (AMSR2 SIC on the day preceding the forecast start date) and the verifying observations. Recall that the goal of the prediction problem is to predict the ground-truth state, not observed SIC, since observations contain random (and probably for SIC systematic) errors. While the systematic errors are much more difficult to address, it should be possible account for random errors in theory by using a different observational product for verification than was used as the predictors in the U-Net model. I realize this might be difficult given the restrictions on resolution for other products, and wanting to use an accurate product, but I would like to see the authors address this in the paper, ideally by using independent obs, but at the minimum raise it as a limitation of the study.

The second is in regard to the Wilcoxon signed-rank statistical test used throughout the study to test the significance of differences in the scores for various models. I'm not familiar with this test, but I was surprised to see that some of the differences were found to be significant, such as at the 1 and 3 day lead times in Fig. 3a,b (but others too). Is there really such little variation in the errors from forecast to forecast that such small differences can be significant, or is there a problem with the test? Maybe the test has problems with autocorrelation in the errors from one week to the next? An alternative option would be a block bootstrap test. Can the authors comment on this concern and are they confident in the results of the test?

Specific minor comments:

Title, abstract, L21, and throughout; I've only ever seen the term "calibration" in statistical post-processing of weather forecasts in the context of probabilistic/ensemble forecasts, in which part of the procedure is an adjustment on the ensemble spread or the shape of the forecast probability distribution. However, I've never seen it for deterministic forecasts like the ones under consideration here. The regression-based approaches to post-process deterministic weather forecasts are known as "model output statistics", but there is no analogous term that I'm aware of yet for deep learning models. To avoid confusion with the probabilistic post-processing literature and methods therein, I think it would be more accurate to replace all instances of "calibration" with simply "post-processing".

L100; Can the authors be more specific when they say "the SIC trend calculated over the 5 days preceding the forecast start date"? What is meant by trend here?

L130; ... "challenging areas" – again some specificity is needed here.

L160; Just to say that I was glad to see this well thought out set of benchmark forecasts used.

Figure 2; The color bar is a bit misleading. Typically differentiating the range of SIC between 95% and 100% is not of any real practical interest, nor is the range between 0% and 15% (although noting the spurious values around 2% SIC in the text maybe noteworthy – typically values less than 15% are just clipped to 0%). Those small variations overwhelm the eye when looking at the maps and make the results look worse than they are. I suggest changing the increment to 5% across the full 0% to 100% range, as it would make any large differences between the maps more evident.

L268 and 269; I would avoid using the word "significant" when describing verification results unless one means "statistically significant". It can be misleading.

L276-277; Does this area of poorer performance in the East Siberian sea have a seasonal component to it (melt vs freeze)? It's a good opportunity to bring up the fact that the use of only one year of test data makes it difficult to say if a feature like this is robust, especially if it's only present in one of the seasons.

---

## Author Comment (AC1)

We would like to thank the reviewer for the constructive comments. Please find below our responses to the comments.

**1 Content**

**The authors Palerme, Lavergne, Rusin, Melsom, Brajard, Kvanum, Sørensen, Bertino and Müller present a study on sea-ice concentration forecasts with lead times up to 10 days. They develop post-processing methods based on supervised machine learning and show that these outperform both TOPAZ4 and persistence sea-ice concentration forecasts. Also, they evaluate the impact of different types of predictors.**

**2 General comments**

**I thank the authors for submitting this already very mature manuscript, which I enjoyed reading and mostly had no problems comprehending. I got a bit lost when it came to the details of the model architecture, but that may be clearer to someone with a stronger AI background. If your target group is people with little AI background, I would suggest boiling down the technical description and spending a bit more time on explaining the terms (see detailed comments below). One aspect where I see room for improvement is the clarity of the motivation. Support of maritime operations is mentioned very prominently in both Abstract and Introduction. Yet, I find little information on crucial aspects for this like timeliness of the forecasts, infrastructure for providing the forecasts to ships operationally or the spatial resolution of the model outcome, compared to what is needed for navigational needs. So, I was wondering whether the current study is intended to be a proof of concept, with the applicability being the focus of future work. It would be good if the authors could clarify this, and elaborate upon for which other aspects their study is relevant besides supporting of maritime operations. In summary, I suggest to accept this paper for publication after minor revisions.**

**3 Specific comments**

**Abstract**

**General impression: A very concise and yet comprehensive description of your paper. I find little to criticise. L1: If operational support is your main motivation, I would suggest to add some brief information on the timeliness of your forecasts to the Abstract, as this is crucial for operational use. By how much does the post-processing delay the availability of forecasts, and does the accuracy increase achieved by post-processing outweigh this?**

We agree with this comment, and we have added the following sentence in the abstract:

*Predicting the sea ice concentration for the next 10 days takes about 4 minutes (including data preparation), which is reasonable in an operational context.*

**Introduction**

**General impression: A very good description of the existing literature on sea-ice forecasting. You could be a bit clearer on what the new aspects are that your study adds to the body of literature. I would also like to see a bit more details on the motivation and relevance of your study. It is certainly relevant, but it could be made clearer for what. Also, you could end the Introduction by formulating your research goals/questions, which can then be picked up in the Discussion and conclusion section.**

**L16: . . . is often limited by their inaccuracies: I agree, but a reference would be good Nevertheless.**

We have added the following reference:

Veland, S., Wagner, P., Bailey, D., Everet, A., Goldstein, M., Hermann, R., Hjort-Larsen, T., Hovelsrud, G., Hughes, N., Kjøl, A., Li, X., Lynch, A., Müller, M., Olsen, J., Palerme, C., Pedersen, J., Rinaldo, Ø., Stephenson, S., and Storelvmo, T.: Knowledge needs in sea ice forecasting for navigation in Svalbard and the High Arctic, Svalbard Strategic Grant, Svalbard Science Forum. NF-rapport 4/2021, 2021

**L17-19: If mentioned in Melsom et al. (2019), consider giving the accuracy for summer instead of a yearly mean, since that is when most of the maritime traffic happens. Large seasonal variability should still be mentioned.**

In Melsom et al. (2019), there is no mean value reported for the summer period, though it can be seen that the errors are larger during the summer in figure 6 from Melsom et al. (2019). We have decided to modify the following sentence:

*Melsom et al. (2019) reported that the location of the ice edge is predicted with a mean accuracy of 39 km in 5-day forecasts from the TOPAZ4 prediction system (Sakov et al., 2012), with large seasonal variability in the forecast performances.*

by:

*Melsom et al. (2019) reported that the location of the ice edge is predicted with a mean accuracy of 39 km in 5-day forecasts from the TOPAZ4 prediction system (Sakov et al., 2012), with larger errors during the summer when most of the maritime traffic occurs (Müller et al., 2023).*

**L22: I find it a bit counter-intuitive to mention seasonal time scales when speaking about weather forecasts, which for me would be up to ten days. I would suggest to either simply speak about forecasts without specification on weather forecasts, or to refer only to those studies which focus on up to ten days lead time.**

The following statement:

*Statistical correction techniques (often called calibration) have been applied to weather forecasts at time scales ranging from hours to seasons*

has been replaced by:

*Statistical correction techniques have been applied to atmospheric forecasts at time scales ranging from hours to seasons*

**Last paragraph: You could be clearer on what precisely the goal of your study is. Which is the gap that you want to fill? Starting the paragraph with a sentence on what the new thing on your study is, and then elaborating on the why, would be a good transition towards the rest of the paper.**

We think that we have already justified this in the beginning of the paragraph, but we have slightly modified this paragraph. Please find below the paragraph in the revised version of the paper:

*Most of the short-term sea ice prediction systems based on machine learning do not use predictors from dynamical sea-ice models (Fritzner et al., 2020; Liu et al., 2021; Grigoryev et al., 2022; Ren et al., 2022; Keller et al., 2023), and it is currently unclear whether adding such predictors would significantly improve forecast accuracy. This study aims at assessing the impact of using predictors from dynamical sea ice models in the development of SIC forecasts from machine learning, as well as the impact of post-processing SIC forecasts from a dynamical sea ice model for lead times from 1 to 10 days.*

**Data**

**General impression: Good to understand. Partly contained elements which I would suggest shifting to the Results or Methods section (see comments below).**

**L78: Good to see that you did this analysis, but I would move it to the Results section and focus on the data description here.**

We agree with the reviewer, and we moved the paragraph which was previously between the lines 71 and 84 to the Results section in a new subsection called "3.1 Sea ice concentration observations".

**L95: Would be interesting to see later on if that impacts the forecast performance.**

We agree that it would be interesting to evaluate the impact of model developments over time. However, though re-forecasts are currently produced at 9 km resolution from ECMWF IFS, they are only produced on Mondays and Thursdays of the current year and on the corresponding date for the previous 10 years. Therefore, if a reforecast dataset is produced on Thursday 25-01-2024, the reforecast dataset will be produced for every 25-01 of the previous 10 years. While only TOPAZ4 forecasts produced on Thursdays are stored in the long term archive, there are no ECMWF reforecasts available for most Thursdays of the preceding 10 years. This makes it difficult to assess the impact of using the operational ECMWF weather forecasts.

**L97–116: Contains many elements where you describe what you do with the data, rather than which data you use. Consider shifting these elements to the Methods section, or merging Data and Methods section.**

We agree with this comment. Therefore, we merged the Data and Methods sections together.

**L97–105: I think it would help to add the categories in the table, and sort the table after categories, in the same order as they appear in the text.**

We have presented the predictors in the text in the same order and the same categories as in table 1. Please find below the new paragraph:

*In this work, the deep learning models have been developed using 8 predictors that can be divided into three categories (table 1). First, two predictors are derived from AMSR2 SIC observations acquired before the forecast start date, and consist of the SIC observations from the day preceding the forecast start date, and the SIC trend calculated over the 5 days preceding the forecast start date (in % per day). The second category consists of weather forecasts from ECMWF that have been averaged between the forecast start date and the predicted lead time. These predictors are the 2-m temperature, as well as the x and y components of the 10-m wind on the grid used for the deep learning models. Then, predictors from the TOPAZ4 ocean model can be considered as the last category. These variables are the SIC forecasts for the predicted lead time, the difference between TOPAZ4 SIC during the first daily time step and the SIC observed the day before (hereafter referred to as "TOPAZ4 initial errors"), and the land sea mask (constant predictor).*

**L113: Does this influence the training, in the sense that the model is trained for points which are discarded later anyway, potentially diminishing the forecast quality for other, "real" points?**

We have tested three different approaches for dealing with land grid points. The first approach consists of considering land grid points as ice-free ocean (the one used in the paper). The second approach consists of interpolating the nearest ocean grid point over land (this approach was used by Wang et al., 2017 and Kvanum et al., 2024). The last approach consists of using partial convolution, which is a method where the land grid points are masked during the convolution operations, and which was used by Durand et al., 2023. We have compared these three approaches in the figure below, and we decided to consider land grid points as ice-free ocean based on this comparison. We have added this figure in the supplement. Furthermore, the use of a land-sea mask in the predictors probably provides enough information for predicting the land grid points correctly.

[Figure]

*Figure S2. Performances of the deep learning models with the Attention Residual U-Net architecture during 2021 (validation period) with three different approaches for filling land grid points. Blue curves: land grid points are considered as ice-free ocean. Black curves: the land grid points are filled using the value of the nearest ocean grid point. Green curves: partial convolution is used, which is a method where land grid points are masked. AMSR2 sea ice concentration observations are used as reference.*

Durand, C., Finn, T. S., Farchi, A., Bocquet, M., and Òlason, E.: Data-driven surrogate modeling of high-resolution sea-ice thickness in the Arctic, EGUsphere [preprint], https://doi.org/10.5194/egusphere-2023-1384, 2023.

Kvanum, A.F., Palerme, C., Müller, M., Rabault, J., Hughes, N. (2024). Developing a deep learning forecasting system for short-term and high-resolution prediction of sea ice concentration, EGUsphere [preprint],

Wang, L., Scott, K., and Clausi, D.: Sea Ice Concentration Estimation during Freeze-Up from SAR Imagery Using a Convolutional Neural Network, Remote Sensing, 9, 408, https://doi.org/10.3390/rs9050408, 2017.

**Methods**

**General impression: Section 3.1 was quite hard to comprehend due to the use of many technical terms, which I myself am not very familiar with. May be different for people with a strong AI background. Sections 3.2 and 3.3 were much clearer to me and do not need improvement, except for one question which I raise below.**

**L124–143: I find the description quite technical. Explaining, or, if they are not crucially needed, omitting terms like "residual connections", "convolutional/attention blocks" or "average pooling" would help to understand non-AI experts to follow your methodology.**

We agree that the description is technical and can be difficult to understand for people who do not have a machine learning background. However, we think that this section is necessary in order to make the work reproducible and we also introduced some novel approaches for sea ice forecasting using deep learning here. Furthermore, we also think that this is not the role of such a paper to explain all the machine learning terms because this would be very long. We rather think that people without a machine learning background interested in understanding the method in depth can read the papers cited in our paper, or / and look at some online tutorials explaining these concepts. It is also worth noting that the rest of the paper can be understood without understanding all the details of the deep learning that we developed. Nevertheless, we have modified this section in order to make it a bit more understandable. Therefore, the following sentences:

*First, some models were developed using residual connections (He et al., 2016) in the convolutional blocks, meaning that the residual was learned at each block. This has been shown to ease neural network training (He et al., 2016). Furthermore, the impact of using attention blocks introduced by Oktay et al. (2018) in the decoder, and designed to improve predictions in challenging areas, is also evaluated. The benefit of using attention blocks for sea ice forecasting was already shown by Ren et al. (2022) who developed an attention block (different from the one used in this study) for sea ice prediction with a fully convolutional network. Finally, average pooling was used in the downsampling blocks of the encoder instead of max pooling due to slightly better performances observed during the tuning phase (see supplement).*

have been replaced by:

*First, some models were developed using residual connections (He et al., 2016) in the convolutional blocks (meaning that the residual was learned at each block), which was shown to ease neural network training (He et al., 2016). It is worth noting that the residual U-Net architecture was used by Keller et al. (2023) for predicting the sea ice extent in the Beaufort*

*sea. Furthermore, the impact of using attention blocks introduced by Oktay et al. (2018) in the decoder, and designed to give more weight (attention) on areas that are challenging to predict (these regions are identified by the attention blocks during training), is also evaluated. The benefit of using attention blocks for sea ice forecasting was already shown by Ren et al. (2022) who developed an attention block (different from the one used in this study) for sea ice prediction with a fully convolutional network. Finally, average pooling was used in the downsampling blocks of the encoder instead of max pooling due to slightly better performances observed during the tuning phase (see supplement).*

**L146: Does this also contain the "fake ocean points" described in L113, and if yes, does that influence the results?**

The land grid points were filled with valid values for oceanic variables. In line 113, we wrote:
*When providing the predictors to the neural networks, all the grid points must contain valid values, meaning that the land grid points must be filled with valid values for oceanic variables*.

In  the manuscript, we wrote the following sentence:

*In order to analyze the full range of SIC values in the forecasts, as well as to strongly penalize large errors, the root mean square error (RMSE) is calculated over all oceanic grid points.*

Because the RMSE is calculated over all oceanic grid points, the land grid points which were filled with valid values were not taken into account for evaluating the deep learning models.

**L167: TOPAZ should be TOPAZ4 (also in other places, occurred several times).**

Thanks for noticing this. We have used TOPAZ4 everywhere in the revised version of the manuscript.

**Results**

**General impression: As for the other sections, I find little to criticise. I did have some open questions, but nothing major.**

**L175: Does the varying number of model parameters influence the results, in the sense that you get more precise results for the Attention Residual U-Net models simply because there is a larger number of model parameters to optimize upon?**

This is a good point. Yes, the number of parameters influences the performances. However, it is difficult to quantify the impact of the number of parameters without changing the architecture. Nevertheless, residual connections are known to speed up model training (less epochs are necessary to achieve similar performances) and attention blocks are designed to improve the

performances in challenging areas (more weight is given to these areas). Therefore, though the impact of the number of parameters is difficult to assess, we believe that residual connections and attention blocks also significantly impact the performances. Furthermore, in figure 3, while the original U-Net architecture is the model containing the lowest number of parameters (31 millions), it performs better than the Attention U-Net (37 millions parameters) for most lead times. This tends to show that the residual and attention blocks have a larger impact on the predictions than the number of parameters.

The following statement:

*It is worth noting that the architecture influences the number of model parameters*

has been replaced by

*It is worth noting that the architecture influences the number of model parameters, which can also influences the performances*

**L184: Did you also see the spurious SIC which the Attention U-Net shows on other dates? Generally, concerning Fig. 2: It is fine to show only one day's maps as example, but it would be good to know if you also looked at other dates and how representative this date is for the overall development, how the RMSE's etc evolve over time. Since the ice refreezes rapidly at this time and the SIC is likely varying less in mid-winter, it would be interesting to see if you can also see that in the temporal development of the RMSE's and the comparison to your benchmarks.**

Thanks for this comment. We have replaced the following sentences:

*The model with the Attention U-Net architecture produces very small positive SIC (lower than 2 %) in large areas where no sea ice is observed during the target date. Nevertheless, it seems that adding residual blocks to this model (resulting in the Attention Residual U-Net architecture) helps to better predict these areas.*

by:

*The model with the Attention U-Net architecture produces very small positive SIC (often lower than 2 %) in large areas where no sea ice is observed during the target date, which is a pattern often observed with this model for other dates as well. Nevertheless, it seems that adding residual blocks to this model (resulting in the Attention Residual U-Net architecture) consistently helps to better predict these areas.*

Furthermore, we consider that the evolution of the RMSE over time is shown in figure 8 of the preprint.

**L200–206: Do you know why the TOPAZ4 RMSE is higher than the persistence benchmarks for all lead times? Wouldn't you expect it to be better? It's not the focus of your paper, so don't spend too much room on this, but it was one of the first things which came to my mind when looking at Fig. 4.**

This is an interesting question, but we do not know why TOPAZ4 performs worse than persistence. We consider that analyzing the reason why TOPAZ4 produces such results is beyond the scope of our paper. Nevertheless, one reason might be that the AMSR2 sea ice concentration product used in our study might significantly differ from the passive microwave observations assimilated in TOPAZ4.

**L206ff: It would be interesting to see how removing predictors influences the runtime of the model. You could discuss, for example, whether the advantage gained in runtime outweighs the small benefit of including TOPAZ4 predictors, and if this might be reason enough to drop the TOPAZ4 predictors altogether.**

We added the following sentences in the section "Discussion and conclusion":

*While it takes less than a second to predict the sea ice concentration for one lead time on a 12 GB GPU (NVIDIA Tesla P100 PCIe) once the list of predictors is available, the full processing chain including the production of the predictors on a common grid takes about 4 minutes for all lead times. This is negligible compared to the time necessary for producing TOPAZ4 forecasts, and therefore reasonable in an operational context.*

Since it takes less than a second to produce the forecasts for one lead time and only 4 minutes for the full processing chain, we consider that the advantage of dropping TOPAZ4 forecasts is very small for the run time of the deep learning model.

**Figure 5: It would be good if the colors of the bars would be consistent with those in Figure 4.**

We made the figures 4, 5, and 8 using consistent colors.

Discussion and conclusion

**The section provides a good summary of your findings, and puts them nicely into perspective with other studies. You could state your conclusions more clearly. This could for example be done by formulating a clear goal or research question(s) at the end of the Introduction which you can pick up and answer here to provide a nice framework for your paper.**

We expressed the question of quantifying the impact of predictors from a dynamical sea ice model on the predictions from the deep learning models in the introduction:

*Most of the short-term sea ice prediction systems based on machine learning do not use predictors from dynamical sea-ice models (Fritzner et al., 2020; Liu et al., 2021; Grigoryev et al., 2022; Ren et al., 2022; Keller et al., 2023), and it is currently unclear whether adding such predictors would significantly improve forecast accuracy. This study aims at assessing the impact of using predictors from dynamical sea ice models in the development of SIC forecasts from machine learning, as well as the impact of post-processing SIC forecasts from a dynamical sea ice model for lead times from 1 to 10 days.*

And we answered this question in the section Discussion and conclusion:

*The impact of using predictors from TOPAZ4 sea ice forecasts is much lower since these predictors lead to a reduction in RMSE of only 2.1 % on average. While the impact of using sea ice forecasts from TOPAZ4 is limited in this study, this does not mean that using predictors from sea ice forecasts does not have stronger potential. TOPAZ4 is an operational system that has been constantly developed since 2012, which can lead to inconsistencies limiting the impact of these predictors. The production of consistent re-forecasts with operational systems could increase the impact of sea ice forecasts in the development of such methods, and should be recommended in the sea ice community. Furthermore, it is likely that more accurate physical-based sea ice forecasts would have larger potential as predictors for machine learning models.*

**Also, I am missing a discussion on the practical applicability of your study. In the Abstract and Introduction, you name increasing marine traffic as one reason why short-term sea-ice forecasting is relevant. It would be interesting to discuss in how far your model is ready to support decision making for marine operations: How fast would your products be available? Do you have the means to transfer them in near-real time? Does the precision of your results meet the requirements of the onboard ship personnel? Or is your paper rather a proof of concept that it is possible to achieve high-quality short-term forecasts using deep learning, and the transfer to near-real time applications is something which could be done in future?**

We agree that this point was not very clear in the preprint. Therefore, we have added the following sentences:

*While it takes less than a second to predict the sea ice concentration for one lead time on a 12 GB GPU (NVIDIA Tesla P100 PCIe) once the list of predictors is available, the full processing chain including the production of the predictors on a common grid takes about 4 minutes for all lead times. This is negligible compared to the time necessary for producing TOPAZ4 forecasts, and therefore reasonable in an operational context. However, the production of TOPAZ4 forecasts will be stopped in February 2024, and the AMSR2 SIC observations used in this study are not available in near real time yet. This prevents the operational use of the post-processing method presented here.*

And the following sentences:

*While this study focused on developing pan-Arctic SIC forecasts at the same resolution as the TOPAZ4 prediction system (12.5 km), there is also a need for higher resolution (kilometer scale) sea ice forecasts (Wagner et al., 2020). This can be addressed by developing regional high resolution prediction systems using deep learning such as the recent works from Keller et al. (2023) and Kvanum et al., (2024).*

---

## Author Comment (AC2)

We would like to thank the reviewer for the constructive comments. Please find below our responses to the comments.

**Review of "Calibration of short-term sea ice concentration forecast using deep learning"**

**Overview:**

**This study by Cyril Palerme and co-authors develops a U-Net deep learning to post-process sea ice concentration forecasts in the Arctic. The model uses predictors from numerical sea ice forecasts, weather forecasts, and satellite sea ice concentration observations, and their sensitivities are examined. Analysis of forecasts over the independent test period of 2022 indicate the deep learning model can outperform several noteworthy benchmark forecasts.**

**General comments:**

**I really enjoyed reading the paper and was encouraged by the results. The daily SIC forecast problem at <10 day lead time is a challenging one, and even state of the art numerical prediction models have a difficult time with it. So, it's encouraging to see how deep learning may be able to help in this regard.**

**Semi-major comments:**

**I have two semi-major comments, one of which might not be possible to address, and the other might not be a problem at all and just my ignorance. The first has to do with the verifying observation choice of the "new" AMSR2 product. I appreciate the analysis done in section 2.1 that compares this product against Norwegian ice charts, and I don't doubt that this is a fine product to use for the kind of the high resolution forecasts being produced. However, at such short lead times, I worry that there is an independence problem between one of the most important predictors in the U-Net model (AMSR2 SIC on the day preceding the forecast start date) and the verifying observations. Recall that the goal of the prediction problem is to predict the ground-truth state, not observed SIC, since observations contain random (and probably for SIC systematic) errors. While the systematic errors are much more difficult to address, it should be possible account for random errors in theory by using a different observational product for verification than was used as the predictors in the U-Net model. I realize this might be difficult given the restrictions on resolution for other products, and wanting to use an accurate product, but I would like to see the authors address this in the paper, ideally by using independent obs, but at the minimum raise it as a limitation of the study.**

We agree with this comment. Our choice of comparing with the ice chart data was motivated by the fact that SAR data is the primary source for the ice chart analysis. In the prioritized list of information sources for ice charts, AMSR2 data ranks as 'h' in a list that runs from 'a' to 'i' (p. 34

in the JCOMM report), so the ice charts are nearly independent of AMSR2 data. The final paragraph in Section 2.1 has been rewritten to the following:

*In addition, the ice charts produced by the Ice Service of the Norwegian Meteorological Institute (https://www.cryo.met.no/en/latest-ice-charts; JCOMM Expert Team on sea ice, 2017) are used as an independent dataset for evaluating the AMSR2 SIC observations and the forecasts developed in this study. The ice charts are manually drawn by ice analysts using several types of remote sensing data. Due to their high spatial resolution, synthetic-aperture radar (SAR) images constitute the main source of information where they are available. Elsewhere, visible and infrared observations are used in priority, while passive microwave retrievals are used where no other observations are available. For evaluating the SIC forecasts, the ice charts were interpolated on the grid used for the deep learning models using nearest neighbor interpolation. It is worth noting that the ice charts provide SIC categories and are not produced during weekends. Therefore, the number of ice charts available in 2022 for evaluating the SIC forecasts varies depending on lead time (between 144 and 243), and is considerably lower than the number of AMSR2 SIC observations available.*

We added the figure below in the paper. This figure represents an evaluation of the ice edge position in the European Arctic using the ice charts from the Norwegian Meteorological Institute as reference.

[Figure]

*Figure 6. Performances of the deep learning models with the Attention Residual U-Net architecture during 2022 (test period) using the ice charts as reference. The ice edge position (defined by the 10 % SIC contour) is evaluated. a) Mean ice edge distance errors depending on lead time. b) Fraction of days in 2022 during which the forecasts from the models with the Attention Residual U-Net architecture outperform the different benchmark forecasts when the forecasts are evaluated using the ice edge distance error. It is worth noting that this evaluation is performed over the area covered by the ice charts from the Norwegian Meteorological Institute (European Arctic), and that the number of forecasts evaluated varies depending on lead time because ice charts are not produced during weekends.*

Furthermore, we have added the following paragraphs for describing this figure:

In the section "3.3 Performances of the deep learning models":

*In order to assess the performances of the SIC forecasts using independent observations, an additional evaluation was performed in the European Arctic using the ice charts from the Norwegian Meteorological Institute as reference (figure 6). Since the ice charts provide sea ice categories (and not SIC as a continuous variable), only the ice edge position is evaluated in figure 6. On average, the forecasts from the deep learning models have an ice edge distance error 40 % lower than TOPAZ4 forecasts, 23 % lower than TOPAZ4 bias corrected, 29 % lower than persistence of AMSR2 SIC, and 22 % lower than persistence of the ice charts. While the forecasts from the deep learning models outperform TOPAZ4, TOPAZ4 bias corrected, and persistence of AMSR2 SIC for all lead times, they have worse performances than persistence of the ice charts for 1-day lead time (the ice edge distance error is 33 % larger). Moreover, only 23 % of the forecasts from the deep learning models outperform persistence of the ice charts for 1-day lead time. Nevertheless, the forecasts from the deep learning models significantly outperform persistence of the ice charts for longer lead times (p-value from the Wilcoxon signed-rank test < 0.05).*

In the section "Discussion and conclusion":

*Using the ice charts from the Norwegian Meteorological Institute as reference, the forecasts from the deep learning models outperform all benchmark forecasts for lead times longer than 1 day in the European Arctic, but are worse than persistence of the ice charts for 1-day lead time. Since the deep learning models are trained using AMSR2 SIC observations for the target variable, it cannot be expected that they perform better than the differences between the two observational products (figure 1). While using ice charts for training deep learning models has been recently proposed by Kvanum et al., 2024, this does not allow to predict the SIC as a continuous variable.*

**The second is in regard to the Wilcoxon signed-rank statistical test used throughout the study to test the significance of differences in the scores for various models. I'm not familiar with this test, but I was surprised to see that some of the differences were found to be significant, such as at the 1 and 3 day lead times in Fig. 3a,b (but others too). Is there really such little variation in the errors from forecast to forecast that such small differences can be significant, or is there a problem with the test? Maybe the test has problems with autocorrelation in the errors from one week to the next? An alternative option would be a block bootstrap test. Can the authors comment on this concern and are they confident in the results of the test?**

Thanks for pointing that out. We realized that we made a couple of mistakes with the interpretation of the Wilcoxon signed-rank test in the preprint. We have modified the following statements:

*These differences are statistically significant (p-value from the Wilcoxon signed-rank test < 0.05) for all lead times and metrics, except for the ice edge distance error for 10-day lead time.*
by:
*These differences are statistically significant (p-value from the Wilcoxon signed-rank test < 0.05) for all lead times and metrics, except for the ice edge distance error for 9-day lead time.*

and

*When comparing the models using all predictors to those developed without TOPAZ4 sea ice forecasts, the differences in RMSE are statistically significant for all lead times, except 10 days.*
by:
*When comparing the models using all predictors to those developed without TOPAZ4 sea ice forecasts, the differences in RMSE are statistically significant for all lead times, except 1 and 10 days.*

Furthermore, the Wilcoxon signed-rank test is an alternative to the Student t-test when the errors are not normally distributed (which is the case for sea ice concentration). The reason why there can be some confusion is that the Wilcoxon signed-rank test does not measure the statistical significance between the mean errors, but between the distribution of the errors. Therefore, it is sometimes possible that the mean errors are relatively close and that the Wilcoxon signed-rank test indicates that the differences are significant. In order to clarify this, we have added the following sentence at the end of section "2.4 Verification scores":

*It is worth noting that the Wilcoxon signed-rank test assesses the statistical significance between the differences in the distribution of the errors (and not between the mean errors).*

**Specific minor comments:**

**Title, abstract, L21, and throughout; I've only ever seen the term "calibration" in statistical post-processing of weather forecasts in the context of probabilistic/ensemble forecasts, in which part of the procedure is an adjustment on the ensemble spread or the shape of the forecast probability distribution. However, I've never seen it for deterministic forecasts like the ones under consideration here. The regression-based approaches to post-process deterministic weather forecasts are known as "model output statistics", but there is no analogous term that I'm aware of yet for deep learning models. To avoid confusion with the probabilistic post-processing literature and methods therein, I think it would be more accurate to replace all instances of "calibration" with simply "post-processing".**

We agree with this comment and we have replaced "calibration" by "post-processing" in the paper. The title of the paper has also been changed, and the new title is "Improving short-term sea ice concentration forecasts using deep learning".

**L100; Can the authors be more specific when they say "the SIC trend calculated over the 5 days preceding the forecast start date"? What is meant by trend here?**

We agree that this statement was not very clear. Therefore, we replaced the following statement:

*"and the SIC trend calculated over the 5 days preceding the forecast start date."*

by:

*"and the SIC trend calculated over the 5 days preceding the forecast start date (in % per day)."*

**L130; … "challenging areas" – again some specificity is needed here.**

We agree with this comment and we have replaced the following sentence:

*Furthermore, the impact of using attention blocks introduced by Oktay et al. (2018) in the decoder, and designed to improve predictions in challenging areas, is also evaluated.*

by:

*Furthermore, the impact of using attention blocks introduced by Oktay et al. (2018) in the decoder, and designed to give more weight (attention) on areas that are challenging to predict (these regions are identified by the attention blocks during training), is also evaluated.*

**L160; Just to say that I was glad to see this well thought out set of benchmark forecasts used.**

Thank you !

**Figure 2; The color bar is a bit misleading. Typically differentiating the range of SIC between 95% and 100% is not of any real practical interest, nor is the range between 0% and 15% (although noting the spurious values around 2% SIC in the text maybe noteworthy – typically values less than 15% are just clipped to 0%). Those small variations overwhelm the eye when looking at the maps and make the results look worse than they are. I suggest changing the increment to 5% across the full 0% to 100% range, as it would make any large differences between the maps more evident.**

We agree with this comment and we have changed the color bar accordingly with increments of 5 % from 0 % to 100 %.

**L268 and 269; I would avoid using the word "significant" when describing verification results unless one means "statistically significant". It can be misleading.**

We agree with this comment, and we have replaced the following sentences:

*Figure 8 shows the seasonal variability in the performances of the deep learning models for lead times of 1, 5, and 10 days. Overall, the calibration shows robust results, with no significant seasonal cycle in the relative improvement compared to TOPAZ4 forecasts and Persistence. Moreover, the deep learning models significantly outperform all the benchmark forecasts for all the months, except in November when the 10-day forecasts are evaluated using the ice edge distance error.*

By:

*Figure 9 shows the seasonal variability in the performances of the deep learning models for lead times of 1, 5, and 10 days. Overall, the deep learning models show robust results, with no clear seasonal cycle in the relative improvement compared to TOPAZ4 forecasts and persistence of AMSR2 SIC. Moreover, the deep learning models outperform all the benchmark forecasts for all the months, except in November when the 10-day forecasts are evaluated using the ice edge distance error.*

**L276-277; Does this area of poorer performance in the East Siberian sea have a seasonal component to it (melt vs freeze)? It's a good opportunity to bring up the fact that the use of only one year of test data makes it difficult to say if a feature like this is robust, especially if it's only present in one of the seasons.**

The forecasts from the deep learning models have better performances compared to TOPAZ4 in the melt season than in the freeze-up season in the East Siberian Sea. We have added the following sentence:

*Nevertheless, it is difficult to determine if these poorer performances in the East Siberian sea are persistent because only one year is used for this analysis.*